

# Null alleles are ubiquitous at microsatellite loci in the Wedge Clam (*Donax trunculus*)

Ciro Rico[1,2,*], Jose Antonio Cuesta[3], Pilar Drake[3], Enrique Macpherson[4], Louis Bernatchez[5] and Amandine D. Marie[1,*]

[1] School of Marine Studies, Molecular Analytics Laboratory (MOANA), Faculty of Science Technology and Environment, The University of the South Pacific, Suva, Fiji
[2] Estación Biológica de Doñana, (EBD, CSIC), Sevilla, Spain
[3] Instituto de Ciencias Marinas de Andalucía (ICMAN, CSIC), Puerto Real (Cádiz), Spain
[4] Centre d'Estudis Avançats de Blanes, (CEAB, CSIC), Blanes, Spain
[5] Institut de Biologie Intégrative et des Systèmes (IBIS), Département de Biologie, Pavillon Charles-Eugène-Marchand, Laval University, Quebec, Canada
[*] These authors contributed equally to this work.

## ABSTRACT

Recent studies have reported an unusually high frequency of nonamplifying alleles at microsatellite loci in bivalves. Null alleles have been associated with heterozygous deficits in many studies. While several studies have tested for its presence using different analytical tools, few have empirically tested for its consequences in estimating population structure and differentiation. We characterised 16 newly developed microsatellite loci and show that null alleles are ubiquitous in the wedge clam, *Donax trunculus*. We carried out several tests to demonstrate that the large heterozygous deficits observed in the newly characterised loci were most likely due to null alleles. We tested the robustness of microsatellite genotyping for population assignment by showing that well-recognised biogeographic regions of the south Atlantic and south Mediterranean coast of Spain harbour genetically different populations.

## INTRODUCTION

Recent studies have reported an unusually high frequency of non-amplifying alleles, also known as null alleles, at microsatellite loci in bivalves (*Hargrove et al., 2015*; *Chiesa et al., 2016*) including the wedge clam (*Nantón et al., 2014*). Null alleles occur when mutations in the binding site of the targeted DNA sequence prevent the efficient annealing of at least one primer resulting in failure of amplification during the PCR reaction. Null alleles can also occur via segmental aneuploidy, where one chromosome has a deletion containing the primer binding site. For instance, in a study of allelic BAC sequences in the Pacific oyster obtained by the Oyster Genome Consortium, 42 of 101 microsatellite loci were found to occur in the hemizygous state, owing to indels of various sizes (P. Gaffney, pers. com., 2016). Null alleles can occur in homozygous state when the samples produce no

Corresponding authors
Ciro Rico, ciro.rico@usp.ac.f
Amandine D. Marie, amandine.marie@usp.ac.fj

amplification at all, and in heterozygous state, when the sample appears as a homozygous individual for a particular locus. *Nantón et al. (2014)*, characterised the first nineteen polymorphic microsatellite markers in *D. trunculus* to assist in the management of the species and allow the delineation of conservation units essential in fisheries. Of the 19 loci characterised in this study, at least 10 showed the presence of null alleles with frequencies ranging from 0.109 to 0.277. Another study aiming to examine the genetic variability and relationships between two Manila clam (*Ruditapes philippinarum*) populations in Korea also suggested the occurrence of null alleles in 7 loci out of 10 (*Kim et al., 2014*). More recently *Chiesa et al. (2016)* demonstrated the occurrence of null alleles through rigorous analytical tests in this species. The presence of null alleles has also been reported in a study that assessed the genetic diversity of four wild and six hatchery stocks of the hard clam (*Mercenaria mercenaria*) in Florida. Null alleles were present in all loci with frequencies ranging from 0.015 to 0.296, but for four loci out of seven, the frequency of null alleles was particularly high (>0.146) (*Hargrove et al., 2015*).

Simulation studies suggest that null alleles with frequencies between 5% and 8% should have only minor effects on classical estimates of population differentiation, but that higher frequencies would bias such parameters (*Chapuis & Estoup, 2007*). Null alleles have been associated with heterozygous deficits in many studies and several have tested their presence and effects in population structure estimates using various analytical and simulation tools (e.g., *Dąbrowski et al., 2015* and references therein). However, to the best of our knowledge no study has empirically tested for its consequences in estimating differentiation using population samples separated by well characterised biogeographic barriers.

The wedge clam constitutes an important fishing resource in southern Spain and Portugal due to its high economic value (*Gaspar, Ferreira & Monteiro, 1999*). In Europe alone, the recorded landings in the last 12 years total 11,202 tons with a maximum yield of 1,355 tons in 2005 (*FAO-FIGIS, 2016*). Overall, the species has experienced a steady decline in recent years reaching only 757 tons in 2014 (*FAO-FIGIS, 2016*). Despite its overexploitation threat, *D. trunculus* has only recently been studied from a population genetics (*Marie et al., 2016*). Understanding connectivity among populations through genetic structure provides tools to determinate the appropriate units and spatial scale for fisheries conservation and management (*Waples & Gaggiotti, 2006*; *Funk et al., 2012*). *D. trunculus* is an Atlantic–Mediterranean warm-temperate species found in the Black Sea, in the Mediterranean Sea (*Bayed & Guillou, 1985*) and from Senegal to the northern Atlantic coast of France (*Tebble, 1966*). It inhabits sandy beaches exposed to tidal rhythms, characterised by intense wave action and sediment instability (*Brown & McLachlan, 1990*). In these environments, populations are capable of reaching sufficiently high densities to support large commercial fleets (*Gaspar, Ferreira & Monteiro, 1999*). Because, wedge clams are filter feeders, they play important roles in the trophic structure of beaches, but can also accumulate xenobiotic compounds making them, ideal model organisms for environmental monitoring (*Saavedra & Bachere, 2006*). For this reason, they may also constitute potential risks for human health when they are eaten and thus have been extensively studied from an ecotoxicology perspective (*Tlili et al., 2010*; *Yawetz et al., 2010*;

*Bouzas et al., 2011*; *Company et al., 2011*; *Hamdani & Soltani-Mazouni, 2011*; *Tlili et al., 2011*).

Motivated by our interest in the conservation genetics of this important shellfish resource, we used sequences generated by one NGS platform (Roche-454) for microsatellite characterisation. Evidence of successful PCR amplification and allelic size variation across a subset of population samples allowed us to optimise 16 loci into four multiplex panels, permitting the PCR amplification of several markers simultaneously. We then genotyped two adjacent locations in the southern Atlantic Coast of Spain. One of them is situated within the boundaries of the Doñana National Park, exploited, due to historic rights, only by artisanal hand-operated dredges with restricted quotas. The second is approximately 85 km apart and is exploited by a commercial boat-operated dredging fleet. We also genotyped a sample from the south Mediterranean Coast of Spain. These samples, located on both sides of the Gibraltar strait, which is a well-known major barrier to gene flow (*Perez-Losada et al., 2007*; *Sá-Pinto et al., 2012*), allowed us to test the strength of the analyses using loci that have high frequencies of null alleles in this species. Finally, we also show through empirical and analytical tests that null alleles are ubiquitous in this species as previously suggested (*Nantón et al., 2014*).

## MATERIALS & METHODS

### Sampling

Samples of *D. trunculus* were collected from three sites, two located in the Atlantic coast and one in the Mediterranean Sea. In the southwest Atlantic Coast of Spain, we collected samples from two adjacent localities, Isla Canela ($n = 66$; 37°10′11.334″N, 7°22′14.25″W) and Doñana National Park ($n = 77$; 36°55′6″N, 6°28′20″W), in the province of Huelva. The third sample was obtained from the Mediterranean Coast of Spain from Caleta de Vélez, Malaga ($n = 52$; 36°44′49″N, 4°4′32″W). The Euclidean geographic distance between the first 2 sites over coast line is 85 km; while between the second and the third site is 287 km (128 km from Doñana National Park to the Strait of Gibraltar and 158 km from there to the third site). The first two samples are separated from the third one by the Strait of Gibraltar which is well-recognised as a major geographical barrier to dispersal among marine taxa (*Perez-Losada et al., 2007*; *Sá-Pinto et al., 2012*). Permit of sampling was approved by the Consejeria de Medio Ambiente (permit number 2011107300001463).

### Library preparation and 454 GS-FLX pyrosequencing

DNA was extracted from different tissues using a slightly modified version of *Aljanabi & Martinez (1997)* salting out method. Namely, after addition of the saline solution, the mixture was centrifuged for 30 min at 10,000 g. Also, DNA precipitation was achieved by incubation at −20 °C with 600 µL isopropanol for 30 min. Then, after washing out the pellet with 70% ethanol, we centrifuged at 10,000 g for 10 min.

For the preparation of the SSR enriched library we followed the protocol of *Santana et al. (2009)*. In brief, we started with 20 µg of high molecular weight genomic DNA (gDNA) to enrich for SSR with ISSR-PCR method (*Zietkiewicz, Rafalski & Labuda, 1994*), but without using the final stage of cloning in a bacterial vector. The primers used were:

ISSR1 (5′-DDB (GTC) 5–3′), ISSR2 (5′-DHB (ATC) 5-3′), ISSR3 (5′YHY (GT) 5G-3′), ISSR4 (5′-HVH (CAT) 5-3′), ISSR5 (5′-NDB (TGT) 7C-3′), ISSR6 (5′-NDV (CT) 8–3′), and ISSR7 (5′-HBDB (AAC) 4–3′). For each trial microsatellite-enriched DNA we analysed in 5 µg in the Pyrosequencing platform Roche 454 GS-FLX. For each sample, a single-lane sequencing run using portioned sections of the PicoTiterPlate was performed according to the manufacturer's protocol preparation. Sample preparation and analytical processing such as base calling, were also performed on site according to the manufacturer's protocol.

## Contig assemblies and microsatellite detection

To uncover sequences containing microsatellites in the pyrosequence reads, we used the mreps software which has been designed to isolate and characterise highly polymorphic microsatellite loci (*Kolpakov, Bana & Kucherov, 2003*). For the purpose of evaluating the number of microsatellite-containing reads from the dataset, we defined two filtering steps. In the first step, we included loci containing di- and tri-, nucleotides SSR with 20 to 60 base pairs of repeated units (i.e., at least 10 tandem repeats for di-, and seven for tri-nucleotides), and a maximum proportion of non-perfect repeats of 0.333.

Since hundreds of potential loci were detected in this filtering step and only loci with sufficiently large flanking sequences around the microsatellite are potentially useful for primer design for successful PCR amplification, we selected loci with a minimum of 40 flanking nucleotides on each side of the tandem repeated units. These loci are referred as "Potentially Amplifiable Loci", or PAL (*Castoe et al., 2012*). From the 85 PAL identified, we selected 52 sequences for subsequent analyses. Primer design was performed using the on-line version of Primer 3 (*Rozen & Skaletsky, 2000*). A total of 52 primer pairs were thus designed and synthesised with the universal M13 primer in the 5′ end of the forward primer for their later analysis using a fluorescently labelled M13 primer according to *Schuelke (2000)*.

## Multiplex optimisation

In the initial development phase, eight individuals were used to verify the amplification of the expected products for the 52 markers. PCR amplifications were performed in a reaction volume of 15 µl, containing 15 ng of gDNA, 0.5 pmol of each primer, 75 µM of each dNTP, 1.5 mM of $MgCl_2$, 1× PCR Green Master Mix buffer (Promega, Madison, WI USA) and 0.5 units of *Taq* polymerase (Promega). The PCR amplifications were done on an Applied Biosystems 9700 DNA thermal cycler using the following conditions: an initial denaturation step at 95 °C for 10 min followed by 35 cycles of 30 s at 94 °C, 1min at either 58 or 50 °C, depending on the primer's melting point, and 1min 30 s at 72 °C, and a last cycle of extension at 72 °C of 10 min. The PCR conditions of successful amplifications were recorded for each locus in order to make the subsequent multiplexing easier. The amplification products were visualized by agarose gel electrophoresis using ethidium bromide staining. A total of 28 primer pairs amplified a clear PCR product and were selected for assessing their allelic size ranges using 48 individuals from the Doñana National Park. The rationale for using 48 samples is that by genotyping many, we aimed to detect a wide spectrum of the genetic diversity present in the population and thus

help in the subsequent design of the multiplex panels. For this, we directly incorporated the fluorescently labelled M13 primer with 6FAM$^{TM}$ (*Schuelke, 2000*), using the same PCR conditions described above. Polymorphism and profile quality were first verified in simplex reactions. The PCR products were then separated on a capillary sequencer (ABI 3130× Genetic Analyzer; Applied Biosystems, Waltham, MA, USA) using GeneScan$^{TM}$ 500 LIZ® Size Standard. Allele sizes were determined with the Gene Mapper® V4.1 program (Applied Biosystems, Waltham, MA, USA). From this analysis, we identified 21 loci that produced clear PCR products with relatively low levels of stuttering that were deemed of sufficient quality for multiplexing. The forward primer of each of these loci was then synthesised again, but this time incorporating in the 5′ end one of the four fluorescent markers 6FAM$^{TM}$, VIC®, NED$^{TM}$ and PET® for multiplex PCR analysis.

We carried out a high number of tests to find optimal multiplexes, grouping several markers of different allelic sizes in a single reaction and adjusting the concentration of the primers according to the strength of the signal. For the multiplex reactions, we used the Qiagen® Multiplex PCR Kit (Qiagen, Toronto, CAN). Final volumes and concentrations of the master mixes were optimised to reduce the total genotyping cost (*Guichoux et al., 2011*). PCR reactions were done in a final volume of 10 μL for all multiplexes, with 3 μL of primer mix, 5 μL of Qiagen Multiplex Mix, and 2 μL of gDNA template (10–15 ng/μL). The cycling conditions for the four multiplexes differed only in the annealing temperature (Table 1): an initial step at 95 °C for 15 min; then 35 cycles at 94 °C for 30 s, 54 to 61 °C for 3 min and 72 °C for 1 min; and a final elongation step at 60 °C for 30 min. Only 16 loci produced codominant fragments (i.e., one allele of maternal and one of paternal origin) in the 48 samples analysed. Table 1 summarises the information about the primers sequences, concentrations in the primer premix (μM), GenBank accession numbers, and multiplex panel annealing temperatures.

## Genotype scoring and analyses

Once multiplex conditions were optimised, we screened a total of 143 individuals from the south Atlantic coast and 52 from the south Mediterranean coast of Spain for variation at 16 polymorphic microsatellite loci (Table 1: 16 loci). Several steps were taken to ensure the consistency and accuracy of the genotype analysis and scoring. An initial allele size spread table was developed from the genotypes of the 195 samples used in the multiplex PCR amplifications for each locus. This table consisted of the observed size ranges for a given allele in each locus, as determined by the GeneMapper® v4.1 (Applied Biosystems). The table was further refined with the same set of samples genotyped in multiplex reactions and then used to manually size each sample according to the scoring range. Afterwards, any sample with a size value beyond the established range was reanalysed. In order to ensure that the allele spread calibration held for each set of samples analysed, we always included the same 6 individuals in each plate to be genotyped as reference standards. All genotype size scores were checked twice to ensure size scoring accuracy. Furthermore, for each marker, clear reading rules were defined and illustrated using screen shots of the chromatograms and provided to a second, experienced scorer to ensure scoring consistency. A first estimate of error rate was obtained by counting mismatches on the

Rico et al. (2017), PeerJ, DOI 10.7717/peerj.3188

**Table 1** **Details of the 16 microsatellite markers developed and optimized for multiplex.** One hundred ninety-five individuals were used for this analysis. Locus identity, type of florescence dye in the forward primer, panel numbers (MP) indicate groupings of loci sharing a multiplex in the PCR reaction, primer sequences, repeat motif SSR, primer concentrations ([C] µM) in the primer mix, number of alleles ($N_A$) found, range of allele sizes (Range, bp), observed ($H_O$) and expected ($H_E$) heterozygosity, $F_{IS}$, departure from Hardy–Weinberg equilibrium (HWE), polymorphism information content (PIC), and GenBank accession numbers. Annealing temperatures for each panel were as follows: Kit 3 = 60° C, Kits 1,2,4 = 58 °C.

| Locus ID | Fluoro | MP | Forward primer | Reverse primer | SSR | [C] µM | $N_A$ | Range | $H_O$ | $H_E$ | $F_{IS}$ | PIC | GenBank Acc. N° |
|---|---|---|---|---|---|---|---|---|---|---|---|---|---|
| D.tru2 | 6-FAM | 4 | ATTCTCCTACGGAGGGGCTA | GCGATGATTTCCTCCGTAAA | (ACA) | 3 | 17 | 97–148 | 0.63 | 0.84 | 0.247 | 0.83 | HG792255 |
| D.tru4 | 6–FAM | 3 | TGCACTTATAATCAACCGGAAG | CTTCCAGCAACACCACGTC | (TG) | 3.5 | 12 | 151–177 | 0.69 | 0.75 | −0.069 | 0.71 | HG792256 |
| D.tru6 | PET | 3 | GTTTTCTCACAGGCGTTCG | GCAGTGATAGGGTTAACGTATTTG | (CA) | 3.5 | 23 | 69–131 | 0.59 | 0.85 | 0.316 | 0.84 | HG792257 |
| D.tru8 | PET | 4 | AATATATTGCAGGCTGGTAGGG | TAAAATTGCCATGCGTGCAG | (ATC) | 3 | 11 | 127–157 | 0.63 | 0.81 | 0.206 | 0.78 | HG792258 |
| D.tru11 | 6-FAM | 2 | AGAACCTGATGTGCTGTGGA | CACGTTAGTACAAAGACCCTTTCC | (GT) | 1 | 21 | 101–145 | 0.57 | 0.79 | 0.268 | 0.78 | HG792259 |
| D.tru14 | NED | 4 | TTTTTGTTCTTCTGAATAGTGCAA | TCGCCATCTTTTGTTGTTGT | CAA) | 3 | 14 | 75–129 | 0.15 | 0.54 | 0.738 | 0.51 | HG792260 |
| D.tru15 | 6-FAM | 1 | TGTCACTAATACAGGATTTCTCACG | AATAGCATCTCTCACACAGACACA | (AC) | 5 | 27 | 219–279 | 0.62 | 0.93 | 0.332 | 0.92 | HG792261 |
| D.tru16 | VIC | 3 | TGCTCCTTATCATTTCAATTGTG | TGCAAACCATCTTCTGGTTG | (AC) | 4.5 | 20 | 70–114 | 0.82 | 0.90 | 0.107 | 0.89 | HG792262 |
| D.tru19 | VIC | 1 | AACACCCATAGCGACGAAAA | GATGACCTGTGAATACATGAAGGA | (AC) | 1.5 | 23 | 126–180 | 0.73 | 0.94 | 0.214 | 0.93 | HG792263 |
| D.tru22 | PET | 1 | TGAAGACATGGCAAAATCCA | TGAGCATATTTCTCTTTCGTAGG | (TTG) | 3 | 8 | 218–245 | 0.27 | 0.69 | 0.601 | 0.64 | HG792264 |
| D.tru23 | NED | 1 | CAAGCACGTTAGACAAAGTCC | ACCTGATGTGTTGTGGACGA | (AC) | 1 | 28 | 105–167 | 0.73 | 0.84 | 0.123 | 0.83 | HG792265 |
| D.tru26 | NED | 2 | TGGAGGTAATTAGATGGTCCAG | ACGCTGGCATCGTTCTCTAT | (AG) | 0.8 | 22 | 57–119 | 0.65 | 0.92 | 0.297 | 0.91 | HG792266 |
| D.tru29 | VIC | 2 | TGAATTTAGTGATTGGCAAAGCTA | ACGGGTGGCATACAACTTGA | (TGT) | 4 | 20 | 262–322 | 0.31 | 0.88 | 0.638 | 0.87 | HG792268 |
| D.tru32 | VIC | 4 | CCGAATGTCCCTTTTGTTGT | TGGGTCCTGGAGGGTAAAAT | (TTG) | 3 | 18 | 196–281 | 0.28 | 0.66 | 0.572 | 0.64 | HG792271 |
| D.tru40 | PET | 1 | GACATTAAGGAGTGGTTGCGTA | CATCAACCGAAAACTCTATAAACTG | (TG) | 1 | 17 | 119–155 | 0.60 | 0.88 | 0.304 | 0.87 | HG792273 |
| D.tru49 | NED | 4 | GAGTATTTCTAACGGTCTTCAAGTTAT | GCATTTATCTTATGTGGTGTTTGC | (CA) | 3 | 11 | 148–182 | 0.24 | 0.78 | 0.663 | 0.74 | HG792275 |

basis of these six positive controls repeated two times in each multiplex panel. Positive controls were essential to include as they allowed the verification of differences across genotyping runs. A second estimate of error rate was obtained by comparing scoring across scorers. Two types of errors were distinguished. Type A corresponds to cases where reader 1 called a genotype as heterozygous and reader 2 typed it as homozygous, or vice versa. Type B corresponds to cases where a different allele was called by either of the scorers (*Castoe et al., 2012*). For each marker, we calculated the total number of alleles ($N_A$), the observed ($H_O$) and expected ($H_E$) heterozygosities, the polymorphism information content (PIC) and $F_{IS}$ using the program CERVUS 3.0 (*Kalinowski, Taper & Marshall, 2007*) while the null-allele frequencies (NAF) were determined using the program ML-NULLFREQ (*Kalinowski & Taper, 2006*). Private alleles, defined as alleles for a given locus present in only one of the two populations (South Atlantic Coast of Spain and South Mediterranean Coast of Spain) were estimated using Convert version 1.31 (*Glaubitz, 2004*). Samples were also tested for linkage disequilibrium and departure from Hardy-Weinberg equilibrium by the Markov chain method using 100,000 dememorization steps with the program Arlequin V 3.5 (*Schneider, Roessli & Excoffier, 2000*; *Excoffier, Laval & Schneider, 2005*). Finally, we investigated whether there was any genetic differentiation between the two Atlantic adjacent localities and subsequently between these and the Mediterranean sample using weighted $F_{ST}$ ($\theta$) statistics by estimating the pairwise fixation index based on allele frequency variation over all loci (*Weir & Cockerham, 1984*). The significance of genetic subdivision was assessed using 1,000 permutations in Arlequin.

We also estimated null allele frequencies and $F_{ST}$ using the software FreeNa (*Chapuis & Estoup, 2007*) with a number of replicates fixed to 25,000. We ran this analysis using the ENA correction method to efficiently correct for the positive bias induced by the presence of null alleles on $F_{ST}$ estimation and so, to provide an accurate estimation of $F_{ST}$. We ran both software programs in order to see if the presence of high frequencies of null alleles can affect the $F_{ST}$ estimates knowing that Arlequin does not take into account the presence of null alleles compared to FreeNa.

## Inbreeding, null allele estimations and genotype artefacts

The possibility of genotyping artefacts was examined in several ways. First, a subset of samples ($N = 94$) were re-amplified for each of the 16 loci under relaxed annealing temperatures (4 °C lower) to verify that additional alleles had not been missed. Second, we used the software MicroChecker (*Van Oosterhout et al., 2004*) to test for stuttering and large allele drop-out and to obtain various estimates of the expected frequency of null alleles (r). These, in turn, were used to calculate the expected number of null homozygotes (i.e., when both alleles remain undetected) using the $4(Nr^2)$ (*Brookfield, 1996*) so that we could compare these estimates to the actual number of individuals that fail to amplify systematically in repeated PCR reactions. The analysis yielded a close match between the expected and observed number of null homozygous. Third, we estimated the frequency of null alleles using ML-NULLFREQ (*Kalinowski & Taper, 2006*) which is a maximum likelihood estimator of the frequency of null alleles in a sample with or without missing data which has been shown to be the single best-performing method (*Dąbrowski et al., 2015*).

Finally, to test the scoring's robustness and power of resolution to depict population genetic structure of these microsatellite loci, we took advantage of the fact that it has extensively been shown, in many marine taxa, that Atlantic populations are often genetically distinct from Mediterranean populations (*Patarnello, Volckaert & Castilho, 2007*). Therefore it is reasonable to hypothesise that the samples from the Atlantic coast separated by 85 km of continuous shoreline would not be genetically distinct from each other, but would be differentiated from the Mediterranean population. To test this hypothesis even when the presence of null alleles in our loci was already confirmed, we took advantage of the algorithm introduced by *Falush, Stephens & Pritchard (2007)* in the program STRUCTURE V2.3.4 (*Hubisz et al., 2009*), which employs a Bayesian clustering method to infer the most likely number of populations ($K$) assuming no a priori structure and allows for the presence of null alleles in the dataset. To test the effects of null alleles in the population structure depicted by these loci, we first estimated the most likely number of populations ($K$) assuming no null alleles and using the LOCPRIOR option and subsequently, we repeated the analysis but incorporating the RECESSIVEALLELES as well as the LOCPRIOR options in STRUCTURE. In this case, the program assumes that the recessive allele is never observed in homozygous state but it might be present. This analysis, partitions multilocus genotypes into clusters, while minimising departure from Hardy–Weinberg expectations and linkage disequilibrium among loci, and it estimates individual ancestry proportions to each putative cluster. We therefore investigated the most likely $K$ running ten independent simulations. All simulations were run using default parameters in the admixture model and with correlated allele frequencies. Each run included 100,000 iterations of burnin, followed by 500,000 MCMC iterations used for parameter estimation. The most likely value of $K$ was chosen using the delta $K$ ($\Delta K$) statistic (*Evanno, Regnaut & Goudet, 2005*), based on the rate of change between successive $K$ values using the STRUCTURE HARVESTER software (*Earl & Von Holdt, 2012*). The algorithm employed by this online program to determine $\Delta K$ by the Evanno method requires that at least three values of sequential $K$ were analysed. Thus, we ran ten independent simulations of $K = 1$–$3$, despite knowing that the maximum number of populations between Atlantic and Mediterranean samples is likely 2. Then, potential admix individuals in each of the proposed clusters were identified using posterior probabilities calculated for each individual in STRUCTURE.

Because of the high frequencies of null alleles, we also ran STRUCTURE with the complete data but removing a different locus at a time for each new run using the same parameters as described above (and including the RECESSIVEALLELES option). This analysis was performed in order to identify the level of information provided by each locus on the level of assignment. Then, we selected the most informative loci and ran again STRUCTURE with and without the RECESSIVEALLELES options still using the same parameters. We selected the most informative loci from the run using the RECESSIVEALLELES option. For that, we created a reference assignment's level from the average level of individuals' assignment of the South-Atlantic Coast of Spain and the South-Mediterranean Coast of Spain respectively. If the average level of individuals' assignment of a run (run when removing a locus) was under the reference one, then, we considered that the locus we removed for this run was very informative.

## RESULTS AND DISCUSSION

A total of 18,795 reads with an average length of 232 bp were obtained from 454 pyrosequencing of a microsatellite enriched library and assembled into 595 contigs. We identified 267 contigs (44.9%) that contained microsatellite loci with at least 10 di- or seven tri-nucleotide repetitions. Of these, there were a total of 85 contigs (31.8%) we identified as PAL of di- or tri-nucleotide microsatellites and 182 contigs (68.2%) as nonPAL (i.e., contigs containing di- or trinucleotide microsatellites without suitable flanking PCR-primer sites). Comparing across the two classes of repeats in PAL sequences, di-nucleotide repeats represented 66.8% and tri-nucleotides 33.2%.

From the microsatellite containing sequences, we selected 52 loci for testing their usefulness in multiplex genotyping and 37 produced PCR amplification when visualised on agarose gels and ethidium bromide staining. Of these 37 loci, nine gave multiple bands, faint bands, or fragments clearly defined but of a different size than the one expected from the original sequence. These loci were discarded from subsequent analyses. Of the remaining 28, 21 could be genotyped and displayed polymorphism in 48 individuals. These 21 loci were deposited in GenBank under accession numbers HG792255.1–HG792275.1. Five loci, D.tru28, 30, 31, 39 and 46, amplified two or three loci simultaneously and were discarded from further analyses. For the remaining loci, the number of alleles per locus ranged from eight to 28 ($\bar{\chi} = 18.25$, $\sigma = 5.86$). Expected and observed heterozygosities, and polymorphic information content ranged from 0.15 to 0.82 ($\bar{\chi} = 0.53$, $\sigma = 0.21$), 0.54 to 0.94 ($\bar{\chi} = 0.81$, $\sigma = 0.11$), and 0.51 to 0.93 ($\bar{\chi} = 0.79$, $\sigma = 0.12$), respectively. $F_{IS}$ estimates ranged from $-0.07$ to 0.74 ($\bar{\chi} = 0.35$, $\sigma = 0.04$) (Table 1). Private alleles were present in all loci for the south Atlantic Coast of Spain and in 11 loci for the South Mediterranean Coast of Spain (Table S1 ). There was no evidence of large allele dropout or genotyping errors due to stutter peaks for the 16 loci. However, all loci but two (D.tru4 and D.tru16), showed a significant excess of homozygotes (deviation from Hardy–Weinberg equilibrium, $P < 0.001$). There was no evidence of linkage disequilibrium between any pair of loci. Genotyping was consistent across the 48 samples, which were each tested twice yielding the same genotypes. The multiplex optimisation yielded a total of four panels for the 16 primer pairs. The numbers of possible multiplex combinations was limited due to the allele size ranges of the markers and differences in the annealing temperatures of the primers. We were thus able to design two 3-plex, and two 5-plex multiplex reactions after excluding the duplicated loci.

Before the advent of next-generation sequencing technologies, no genetic marker has found such widespread use as microsatellites in the last two decades. This study joins an ever increasing number of others that have successfully employed NGS to detect a virtually unlimited number of polymorphic microsatellites across a wide variety of taxa (e.g., (Wang et al., 2012b; Wasimuddin et al., 2012; Whitney & Karl, 2012; Miller et al., 2013). Our study takes advantage of Roche-454 titanium chemistry. Microsatellite-containing sequences from shotgun and enriched libraries have also been reported across a broad range of taxa in recent years (Angeloni et al., 2011; Santure et al., 2011; Wang et al., 2012a; Wang et al., 2012b; Zhang et al., 2012). Together, these studies demonstrate that the main

obstacle of costly *de novo* isolation for using microsatellite markers for population studies and characterisation has been rapidly overcome by the advent of NGS technologies. More importantly, however, is the fact that a growing proportion of research projects, working on modest financial resources, are now capable of developing a suitable number of microsatellite markers at minimal laboratory costs for virtually any taxa (*Schoebel et al., 2013*). The number of microsatellite loci detected in this study illustrates how abundant these sequences are in eukaryotic genomes. Comparing the number of loci detected across studies and taxa is inherently difficult because genomes vary substantially in their frequency of microsatellites and genome size (*Schoebel et al., 2013*). However, the number of loci detected here favourably compares with other studies (e.g., *Wang et al., 2012a*; *Wang et al., 2012b*; *Zhang et al., 2012*). For example, in a recent study that used this technology, *Schoebel et al. (2013)* carried an extensive survey of microsatellite abundance in 17 non-model species including plants, fungi, invertebrates, birds and a mammals and determined if flanking regions were suitable for primer development. They concluded that that depending on the species, a different amount of 454 pyrosequencing data might be required for successful identification of a sufficient number of microsatellite markers for ecological genetic studies. Irrespective of the difficulty to compare microsatellite abundance in different taxa, an important message of our study is that a virtually unlimited number of microsatellite loci can be identified from the large amount of sequence data generated with NGS technologies. It is worth noticing as well that 454 pyrosequencing has been replaced by other technologies which generally yield many more sequences, but with shorter reads. The implications of these shorter reads may have an important impact on marker development as microsatellites require the availability of long flanking sequences for primer design. However, we have to consider that, despite the many advantages of microsatellite markers, admittedly, the development of SNP markers by NGS or SNP genotyping by sequencing have increasingly become attractive alternatives for many reasons such as availability of high numbers of annotated markers, low-scoring error rates, improved genotyping results for poor quality samples, highly informative in terms of segregation among populations, and the ability to examine neutral variation and regions under selection, among others (*Brumfield et al., 2003*; *Rosenberg et al., 2003*; *Morin et al,, 2004* ; *Liu et al., 2005*; *Grewe et al., 2015*).

Once the multiplex panels were optimised, we genotyped a total of 195 individuals from the three sampled locations. Table 1 summarises the genetic diversity found in each locus for all the samples analysed. The initial uncorrected typing error rate was 0.0023 per reaction or 0.0015 per allele. The frequency of single-locus genotypes missing in the data set was <0.05. Adjacent allele scoring error analysis of the final corrected data set yielded no significant deviations ($X^2 < 2$, $P > 0.15$), hinting that there were very few remaining scoring errors. From individuals genotyped as controls in each plate, we calculated a mistyping rate of 0.0019 per allele genotyped. Type A and type B error rates were 0.015 and 0.021 for multiplex 1, 0.013 and 0.008 for multiplex 2, 0.026 and 0.051 for multiplex 3, and 0.009 and 0.014 for multiplex 4, and 0.0152 and 0.023 across all loci and were mainly caused by excessive stuttering of some long alleles. Once again, all loci but D.tru4 and D.tru16 showed a significant excess of homozygotes (deviation from Hardy–Weinberg equilibrium,

**Table 2  Characteristics of the 10 microsatellite markers with original (OA) and new (NA) annealing temperature for 94 individuals (Isla Canela: 27; Doñana: 37; Caleta de Vélez: 30) including the number of alleles ($N_A$), range of allele sizes (Range, bp), observed ($H_O$) and expected ($H_E$) heterozygosity, polymorphism information content (PIC) and null-allele frequency (NAF).**

| Locus | $N_A$ | | Range | | $H_O$ | | $H_E$ | | PIC | | NAF | |
|---|---|---|---|---|---|---|---|---|---|---|---|---|
| | OA | NA | OA | NA | OA | NA | OA | NA | OA | NA | OA | NA |
| D.tru4 | 11 | 34 | 151–179 | 99–209 | 0.641 | 0.809 | 0.759 | 0.930 | 0.718 | 0.920 | 0.093 | 0.065 |
| D.tru6 | 18 | 26 | 71–131 | 69–161 | 0.527 | 0.923 | 0.842 | 0.882 | 0.821 | 0.868 | 0.239 | −0.030 |
| D.tru11 | 20 | 19 | 101–145 | 111–149 | 0.581 | 0.723 | 0.807 | 0.820 | 0.792 | 0.801 | 0.173 | 0.056 |
| D.tru15 | 25 | 23 | 219–279 | 219–281 | 0.670 | 0.759 | 0.928 | 0.936 | 0.918 | 0.926 | 0.158 | 0.101 |
| D.tru19 | 22 | 20 | 126–176 | 128–170 | 0.734 | 0.841 | 0.937 | 0.937 | 0.928 | 0.927 | 0.120 | 0.052 |
| D.tru22 | 7 | 8 | 221–245 | 215–245 | 0.233 | 0.333 | 0.658 | 0.692 | 0.594 | 0.636 | 0.480 | 0.342 |
| D.tru23 | 22 | 22 | 107–159 | 107–151 | 0.702 | 0.886 | 0.838 | 0.864 | 0.823 | 0.850 | 0.093 | −0.020 |
| D.tru26 | 19 | 21 | 57–103 | 67–111 | 0.696 | 0.783 | 0.924 | 0.934 | 0.913 | 0.925 | 0.140 | 0.086 |
| D.tru29 | 17 | 13 | 268–322 | 265–310 | 0.344 | 0.203 | 0.872 | 0.851 | 0.855 | 0.828 | 0.434 | 0.611 |
| D.tru40 | 17 | 18 | 119–163 | 121–161 | 0.596 | 0.830 | 0.866 | 0.885 | 0.849 | 0.869 | 0.182 | 0.030 |

$P < 0.001$) (Table 1). Exact tests for genotypic linkage disequilibrium confirmed the absence of physical linkage among most loci ($P < 0.05$ after Bonferroni correction).

The results of the re-amplification of a subsample of individuals under lower stringency mostly confirmed the above findings. However, only 10 loci produced codominant fragments (i.e., one allele of maternal and one of paternal origin) under relaxed annealing temperatures (Table S1 : Original Annealing Temperature, New Annealing Temperature). Locus D.tru16, which initially had not deviated from HWE, amplified multiple loci and thus could not be used for this comparison. Furthermore, two additional loci showed null allele frequencies not different from zero (D.tru6 and D.tru23). This result suggests that non-amplifying alleles can be detected by reducing the stringency of the annealing temperature. For the remaining eight loci, no differences were observed in the heterozygous deficit or the null allele frequencies supporting the presence of null alleles in the wedge clam genome (Table 2).

The fact that most loci had large heterozygous deficits was an intriguing result which deserves a thorough discussion. The first consideration that needs to be evaluated in addition of the occurrence of null alleles, is the presence of large allele dropout. Another consideration is the level of stuttering as it may hamper the reliable scoring of alleles differing within one or two repeated units in some loci. Another possible explanation could be the differences in allele size as reported by previous studies that have found that loci with longer alleles tend to have higher dropout rates than those with shorter alleles (*Sefc et al., 2003*; *Buchan et al., 2005*; *Broquet, Ménard & Petit, 2007*). In our study, allele size ranges ranged from 60 to 24 ($\bar{x} = 46.2$, $\sigma = 12.12$) and from 110 to 30 ($\bar{x} = 54.7$, $\sigma = 26$) for the original and relaxed annealing temperatures respectively (Table 2). It is thus expected that the largest allele at a locus would amplify with less efficiency than the shortest one as the latter would outcompete the former during amplifications (*Sefc et al., 2003*; *Buchan et al., 2005*; *Broquet, Ménard & Petit, 2007*). This will artificially increase the frequency of null alleles even in the absence of true null alleles.

The analysis of null allele frequencies per population using the maximum likelihood estimates implemented in ML-NullFreq confirmed the presence of null alleles in all loci but D.tru4 and D.tru16. For these loci, null-allele frequencies estimates ranged between 0% and 31.2% ($\bar{\chi} = 0.160$, $\sigma = 0.098$) and all loci but D.tru4, D.tru16 and D.tru23 had a null-allele frequencies above 5% (Table 3). Results from FreeNa showed lower null allele frequencies for each locus (except for the locus D.tru19) than results obtained using ML-NullFreq. Null allele frequencies estimates ranged between 0% and 29.3% ($\bar{\chi} = 0.149$, $\sigma = 0.090$; Table 3). This high frequency of null alleles found in this study is consistent with many reports in other bivalve species (*Launey et al., 2002*; *Nantón et al., 2014*; *Chiesa et al., 2016*). Clams are filter feeders and thus, are prone to accumulate xenobiotic compounds (*Tlili et al., 2010*; *Yawetz et al., 2010*; *Bouzas et al., 2011*; *Company et al., 2011*; *Hamdani & Soltani-Mazouni, 2011*; *Tlili et al., 2011*). It is plausible that the mutagenic action of some pollutants such as heavy metals (e.g., mercury or cadmium) (*Wong, 1988*) can be responsible for an increased mutation rate in clams. This could be further investigated by comparing samples representing genetically distinct populations where organisms are either heavily exposed to pollutants in beaches adjacent to major urban areas or those occupying habitats in remote locations where the level of pollution is likely to be low and far from industrial pollution.

As expected due to the absence of biogeographic barriers and the proximity of the sampling sites in the south Atlantic coast of Spain, the fixation index between samples from Isla Canela and Doñana National Park was not significantly different from zero (Arlequin: $F_{ST} = 0.0028$–$P > 0.05$; FreeNa: $F_{ST}$ including ENA correction $= 0.0025$) indicating uninterrupted migration of larvae across the area. Furthermore, as predicted, the fixation index between these two localities and the Mediterranean sample was significantly different from zero (Arlequin: $F_{ST} = 0.033$–$P < 0.001$; FreeNa: $F_{ST}$ including ENA correction $= 0.032$). The STRUCTURE analysis found that the two populations from the south Atlantic coast of Spain were genetically homogeneous while they were significantly different from the Mediterranean population with the highest $\Delta K$ value equal to the number of biogeographic regions ($K = 2$) and the programme showed that the 10 independent runs from $K = 1$ to $K = 3$ produced consistent results whether or not the RECESSIVEALLELES option was used in the analysis (Fig. 1). Calculation of the statistic $\Delta K$ from the STRUCTURE runs indicated that the two biogeographic regions analysed have clearly differentiated populations.

When running STRUCTURE with the complete data set without a different locus at a time for each run, our results also showed that three microsatellite loci were very informative (D.tru4, D.tru29 and D.tru32; Fig. S1, Table S2 ). Indeed, compared to the reference average level of individuals' assignment (84.7% $\pm$ 2.5 and 90.6% $\pm$ 1.7 for the South Atlantic Coast of Spain and the South Mediterranean Coast of Spain, respectively), without these three loci, the average level of individuals' assignment dropped to 65.5% $\pm$ 6.5, 68.2% $\pm$ 3.7 and 77.9% $\pm$ 2.6 in the South Atlantic Coast of Spain and to 38.2% $\pm$ 5.4, 89.5% $\pm$ 2.2 and 89.9% $\pm$ 2.8 in South Mediterranean Coast of Spain, respectively, when excluding either locus D.tru4, D.tru29 or D.tru32. Surprisingly, two of these loci (D.tru29 and D.tru32) have large frequencies of null alleles (0.311 $\pm$ 0.030 and 0.228 $\pm$ 0.027 for the locus D.tru29 and D.tru32 respectively; Table 3) confirming that even when microsatellite loci display

Rico et al. (2017), *PeerJ*, DOI 10.7717/peerj.3188

**Table 3 Estimate of null allele frequencies per locus and populations.** The maximum likelihood and the ENA correction methods were used, using ML-NullFreq and FreeNa respectively.

| Locus ID | Isla Canela | | Doñana | | Caleta de Vélez | | Mean | | SD | |
|---|---|---|---|---|---|---|---|---|---|---|
| | ML-NullFreq | FreeNa | ML-NullFreq | FreeNa | ML-NullFreq | FreeNa | ML-NullFreq | FreeNa | ML-NullFreq | FreeNa |
| D.tru2 | 0.168 | 0.168 | 0.106 | 0.106 | 0.107 | 0.09 | 0.127 | 0.121 | 0.036 | 0.041 |
| D.tru4 | 0 | 0 | 0 | 0 | 0 | 0 | 0 | 0 | 0 | 0 |
| D.tru6 | 0.106 | 0.106 | 0.112 | 0.112 | 0.222 | 0.173 | 0.147 | 0.131 | 0.065 | 0.037 |
| D.tru8 | 0.144 | 0.114 | 0.125 | 0.125 | 0.05 | 0.044 | 0.106 | 0.095 | 0.05 | 0.044 |
| D.tru11 | 0.113 | 0.113 | 0.139 | 0.117 | 0.084 | 0.084 | 0.112 | 0.105 | 0.028 | 0.018 |
| D.tru14 | 0.307 | 0.264 | 0.356 | 0.238 | 0.273 | 0.273 | 0.312 | 0.259 | 0.042 | 0.018 |
| D.tru15 | 0.199 | 0.16 | 0.189 | 0.164 | 0.164 | 0.142 | 0.184 | 0.155 | 0.018 | 0.011 |
| D.tru16 | 0.014 | 0.014 | 0.033 | 0.033 | 0.083 | 0.083 | 0.043 | 0.043 | 0.036 | 0.036 |
| D.tru19 | 0.088 | 0.088 | 0.111 | 0.111 | 0.102 | 0.102 | 0.1 | 0.101 | 0.012 | 0.012 |
| D.tru22 | 0.3 | 0.282 | 0.263 | 0.248 | 0.23 | 0.204 | 0.264 | 0.245 | 0.035 | 0.039 |
| D.tru23 | 0.038 | 0.038 | 0.064 | 0.064 | 0.008 | 0.008 | 0.037 | 0.037 | 0.028 | 0.028 |
| D.tru26 | 0.09 | 0.09 | 0.165 | 0.165 | 0.204 | 0.165 | 0.153 | 0.14 | 0.058 | 0.043 |
| D.tru29 | 0.277 | 0.277 | 0.322 | 0.322 | 0.333 | 0.28 | 0.311 | 0.293 | 0.03 | 0.025 |
| D.tru32 | 0.258 | 0.258 | 0.219 | 0.219 | 0.206 | 0.206 | 0.228 | 0.228 | 0.027 | 0.027 |
| D.tru40 | 0.103 | 0.103 | 0.223 | 0.223 | 0.09 | 0.09 | 0.139 | 0.139 | 0.073 | 0.073 |
| D.tru49 | 0.299 | 0.299 | 0.359 | 0.335 | 0.222 | 0.222 | 0.293 | 0.285 | 0.069 | 0.058 |

**Notes.**

SD, standard deviation.

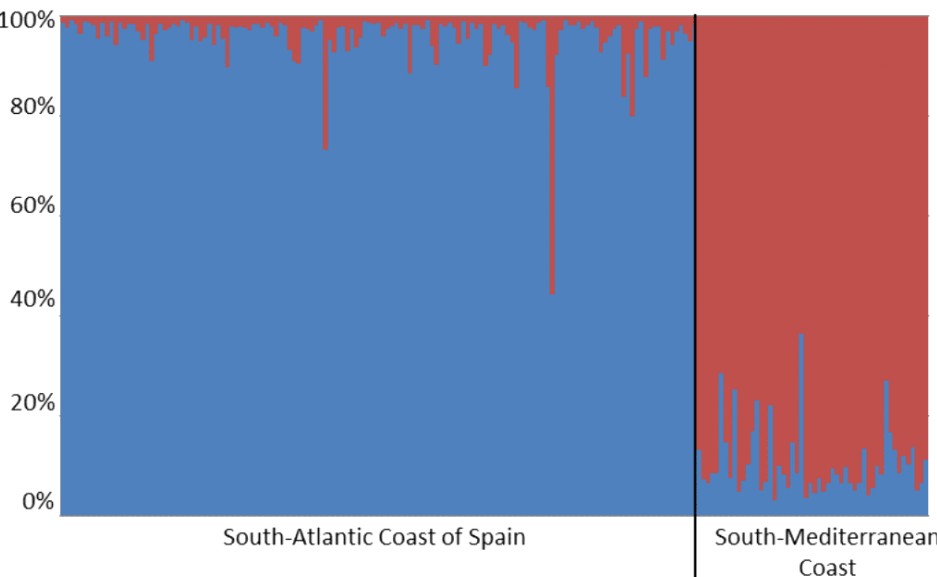

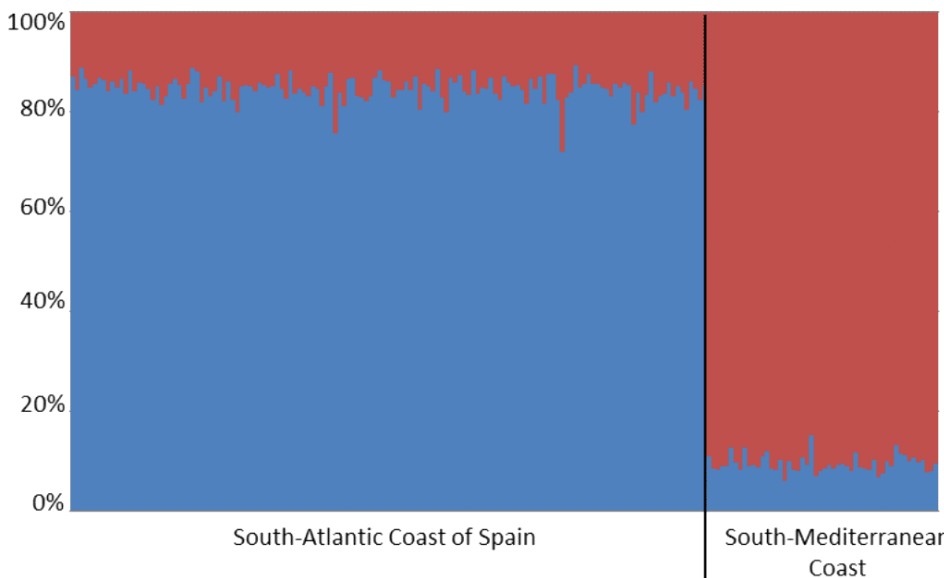

**Figure 1** **Clustering analysis in STRUCTURE considering information about population of origin** ($k = 2$). Individuals are represented as vertical bars, where the amount of each colour indicates the proportion of each inferred cluster. A represents the clusters obtained in the simulations considering the absence of null alleles while B represents those assuming them. Isla Canela and Doñana are represented by the label "South-Atlantic Coast of Spain", and Caleta de Vélez by the label "South-Mediterranean Coast".

a large frequencies of null alleles, they can, nevertheless, be useful to identify the genetic structure of populations. This conclusion was also corroborated with the $F_{ST}$ results using FreeNa and Arlequin, which showed that the use of the null allele's correction option had no effect in the $F_{ST}$ estimates.

To the best of our knowledge, it is unusual that 87.5% of newly developed loci are affected by any form of genotyping errors or allele dropout during PCR due to the presence of long alleles. Although, simulation studies suggest that null alleles with frequencies between 5% and 8% should have only minor effects on classical estimates of population differentiation (*Chapuis & Estoup, 2007*), in our case null allele frequencies will render all these loci, but three, completely useless in population genetic analyses if the heterozygous deficits were due to null alleles. The fact that we found panmixia between samples from adjacent localities with no apparent barriers to gene flow and a well-defined structure for populations separated by a well-characterised biogeographic barrier suggests that the large heterozygous deficits we found, cannot be entirely due to null alleles or genotyping artefacts. If this was the case, one would have expected significant, albeit artificial, genetic differences between the two adjacent localities. Alternatively, it suggests that despite the presence of null alleles, clearly differentiated populations can be detected, although levels of genetic diversity are most likely underestimated. Admittedly, however, results pertaining to genetic structure presented here should be interpreted cautiously until they are corroborated by other approaches, namely SNP genotyping.

## CONCLUSIONS

In conclusion, we have shown that null alleles at microsatellite loci are unusually abundant in the wedge clam and that genotyping artefacts are unlikely to completely explain the large heterozygous deficits observed in all population samples. We have also shown that the high frequencies of null alleles observed at these loci do not appear to have a significant effect in the population genetic parameters commonly assessed for microsatellite loci. The expected population structure between Atlantic and Mediterranean samples was confirmed for the tests carried out. Furthermore, the unusually high frequency of null alleles reported in the wedge clam and in other bivalve species is worth investigating further. As mentioned above, it is reasonable to hypothesise that pollutants may increase mutation rates in bivalves, resulting in high frequencies of null allele. Considering the economic value of *D. trunculus*, additional investigations are essential to validate this assumption.

## ACKNOWLEDGEMENTS

We are grateful for the support given, throughout the project, by the Coordination Office of Donana National Park and the Consejeria de Agricultura, Pesca y Desarrollo Rural of the Andalusian Government. CR also acknowledges MI Rico, F Hiraldo Cano, JJ Negro Balmaseda, E Normandeau and G Côté for their support during his stay in Canada. We gratefully acknowledge the Associate Editor and the two referees, Mariah Scott and Patrick Gaffney, for their relevant comments which allowed us to considerably improve the manuscript.

### Funding

CR was supported by the Spanish Ministry of Education (Grant no PR2010-0601) and the Spanish Scientific Research Council (CSIC). This research was financially supported by Organismo Autonomo de Parque Nacionales Grant OAPN122_2010 to CR, EM, JAC and PD. ADM was supported by the Strategic Research Themes of the University of the South Pacific. The funders had no role in study design, data collection and analysis, decision to publish, or preparation of the manuscript.

### Grant Disclosures

The following grant information was disclosed by the authors:
Spanish Ministry of Education: PR2010-0601.
Spanish Scientific Research Council (CSIC).
Organismo Autonomo de Parque Nacionales Grant: OAPN122_2010.
Strategic Research Themes of the University of the South Pacific.

### Competing Interests

The authors declare there are no competing interests.

### Author Contributions

- Ciro Rico conceived and designed the experiments, performed the experiments, analyzed the data, wrote the paper, reviewed drafts of the paper.
- Jose Antonio Cuesta, Pilar Drake and Enrique Macpherson contributed reagents/materials/analysis tools, wrote the paper, reviewed drafts of the paper.
- Louis Bernatchez contributed reagents/materials/analysis tools, wrote the paper, reviewed drafts of the paper, laboratory and sequencing facilities.
- Amandine D. Marie analyzed the data, wrote the paper, prepared figures and/or tables, reviewed drafts of the paper.

### Field Study Permissions

The following information was supplied relating to field study approvals (i.e., approving body and any reference numbers):

Permit of sampling was approved by the Consejeria de Medio Ambiente (permit number 2011107300001463).

### DNA Deposition

The following information was supplied regarding the deposition of DNA sequences:

The locus described here are accessible via GenBank accession numbers: HG792255, HG792256, HG792257, HG792258, HG792259, HG792260, HG792261, HG792262, HG792263, HG792264, HG792265, HG792266, HG792268, HG792271, HG792273, HG792275.

### Data Availability

The raw data has been supplied as a Supplementary File.

## Supplemental Information

Supplemental information for this article can be found online at http://dx.doi.org/10.7717/peerj.3188#supplemental-information.

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
