# Peer review of "Null alleles are ubiquitous at microsatellite loci in the Wedge Clam (Donax trunculus)"

_PeerJ, doi:10.7717/peerj.3188_

## Round 0.1 · original submission · Major Revisions

I now have responses from 2 referees who both agree that the manuscript has value, but both also point out that the manuscript could be further developed to consider additional explanations for the observed results. While the revisions requested by each referee are relatively straightforward, they are extensive enough and request a softening of the conclusions that these reviews merit a major revision. I do not expect that there is anything you cannot deal with in these comments however, and I expect that a conscientious revision will ultimately result in a publishable submission. I look forward to seeing your revised manuscript.

·

Basic reporting

This is a well written paper, with a clear and logical story line throughout. There are small grammatical and spelling errors throughout the paper, which I commented on below. Most of the errors relate to proper tenses, run on sentences, and comma use. However, these are simple corrections to make and I do not feel that they tarnish the importance of this paper overall.

I am not sure why, but the line numbering in the pdf is different than that in the Microsoft Word Document. The numbering below is based off of the pdf document. Also, I made changes based on spelling used in the States. If this is incorrect, simply ignore those suggestions.

Abstract: The abstract provided a clear overview of the research. It was very concise.
Small Abstract Errors:

25 wedge clam Donax trunculus> wedge clam, Donax trunculus

25 several test > several tests

27 that well recognised (spelling error) > that the well-recognized
Please check the rest of the document for this error

Introduction: The introduction is well referenced and the authors provided a clear context for their research. While the authors touched on the economic importance of the species, I feel that more emphasis is needed on the importance of understanding its population-level genetic composition. The last paragraph has a lot of information in it that are pieces of the methods section. Please revise the last paragraph to focus on the purpose and not the details of the individual research steps.
Small Introduction Errors:

56-61 This sentence is very long and can be improved by being split into multiple sentences

59 none has empirically> none have empirically

63 Only in Europe, > “In Europe,” or “In Europe alone,”
(The initial word disrupts the sentence flow)

64 tons in 2005 from where, it has experienced a steady decline reaching > tons in 2005 (FAO-FIGIS 2013). Overall, the species has experienced a steady decline, reaching
(If this sentence should be saying that the decline happened after 2005, please change it accordingly)

68 tidal rhythms characterised (spelling error)> tidal rhythms, characterized
Please check the rest of the document for this error

82 Why were di- and tri- repeats preferred?

88-92 This sentence is very long and can be improved by being split into multiple sentences

94 When you reference the Gibraltar strait in the rest of the paper, you describe it as a known barrier you are using to test the strength of the analyses using loci with many null alleles. In this line, you talk about testing the strength of the strait as a geographic barrier. Please correct this.

95 trough> through

Methods: The methods section had a strong logical flow and contained all necessary information. There were several instances of reporting results in the methods section, but these instances could be considered methods since they were part of the research’s development process. This concern is mainly a difference of style between myself and the authors. Also, perhaps I missed this, but can you describe how many samples you got from each site?

Paragraph 1 of the Methods section: You begin the paragraph by describing two populations and then numerically describe three sites. This mismatch of labels makes the beginning of your methods section muddled. Please rephrase the sentences to describe the two Atlantic sites as the Isla Canela and Doñana National Park. Then describe the second, Mediterranean population as the Caleta de Vélez samples. Please double check the rest of your paper to make sure that these labels stay consistent.

129 should “mreps” be captilized?

168 synthesised (spelling) again but this time incorporating in the > synthesized again, but this time we incorporated the
(please make sure I retained the meaning of the original sentence, also check the rest of the document for the spelling error)

177 annealing temperature: an initial > annealing temperature (Table 1): an initial
(I feel this makes the location of this information clearer).

189 Do you have a version of GeneMapper that you used and a software citation?

196-199 to a second experienced scorer to ensure scoring consistency. A first estimate of error rate was obtained by counting mismatches on the basis of these six positive controls repeated 2 times in each multiplex panel. Positive controls are essential to include as they allow the >
to a second, experienced scorer to ensure scoring consistency. A first estimate of error rate was obtained by counting mismatches on the basis of these six positive controls repeated 2 times in each multiplex panel. Positive controls were essential to include as they allowed the

216 artefacts > artifacts
Please check the rest of the document for this error

232-233 hypothesise (spelling) that the samples from the Atlantic coast separated by 85 km of continuous shoreline would not be genetically different> hypothesize that the samples from the Atlantic coast separated by 85 km of continuous shoreline would not be genetically distinct
(they will likely be different, but only to a small degree)
253 run>ran

254 samples is 2 > samples is likely 2

Results and Discussion: You laid out your findings in an interesting and logical way. You also provided context for them through comparisons to other research. I would like more elaboration on how the unique nature of this situation may have played a role in the null alleles not impacting the separation of the two populations, such as the strong geographic barrier or ancient split between the two groupings. These findings are interesting, but need to admit potential limitations. Please go through this section and check that the writing remains in past tense.

Small Results and Discussion Errors:
269-270 gave multiple or faint bands or fragments clearly defined > gave multiple bands, faint bands, or fragments clearly defined

273 number > numbers

278- 284 peaks for 16 loci. However, all loci but two (D.tru4 and D.tru16) showed a significant excess of homozygotes (deviation from Hardy– Weinberg equilibrium, P< 0.001). There was no evidence of linkage disequilibrium between any pair of loci. Genotyping was consistent across the 48 samples, which were tested twice which yielded the same genotypes. The multiplex optimization (spelling) yielded a total of 4 panels for the 16 primer pairs. The numbers of possible multiplex combinations was limited due to the allele size ranges of the markers and because, we had to take into account the annealing temperatures >
peaks for the 16 loci. However, all loci, but two (D.tru4 and D.tru16), showed a significant excess of homozygotes (deviation from Hardy– Weinberg equilibrium, P< 0.001). There was no evidence of linkage disequilibrium between any pair of loci. Genotyping was consistent across the 48 samples, which were each tested twice yielding the same genotypes. The multiplex optimization yielded a total of 4 panels for the 16 primer pairs. The numbers of possible multiplex combinations was limited due to the allele size ranges of the markers and differences in the annealing temperatures of the primers.

298-300 research projects working on modest financial resources are now capable of developing a suitable number of microsatellite markers at minimal laboratory costs and for virtually any taxa>
research projects, working on modest financial resources, are now capable of developing a suitable number of microsatellite markers at minimal laboratory costs for virtually any taxa

315 sequences but with shorter reads. The implications of this shorter reads> sequences, but with shorter reads. The implications of these shorter reads

335 Locus D.tru16 which initially did not deviated from HWE> Locus D.tru16, which initially had not deviated from HWE,

340 frequencies confirming the presence > frequencies supporting the presence

342 deficits is indeed a very intriguing > deficits was an intriguing

343-346 This sentence is very long and can be improved by being split into multiple sentences

348-349 is expected that the more different alleles are in size the more likely one would outcompete>
is expected that the more unique alleles are in size (?) the more likely one would outcompete
(I am not sure what you mean by this sentence. I cannot explain it more thoroughly than that)

377 barrier point out in the direction that > barrier suggests that

379 significant albeit artefactual (spelling, in US and UK) genetic differences > significant, albeit artificial, genetic differences

Conclusion: The conclusion section wrapped up the paper well, without being repetitive. However, you speculate on a whole new concept in the conclusion. While an interesting idea, you need to develop it in the results and discussion section before you can mention it in the conclusion.

Small errors in the conclusion:
391 species is worth of further studies> species is worth further study

Figures and Tables:
Your figures and tables are clear, with well written legends. I would prefer to see the variable names and descriptions as part of their respective table legend (581-582 and 587-588). As legends should be fully informative, avoid abbreviations, such as MP.

Small errors in the table
Table 1: I am assuming the “Range” is in bp, but please state that

576: MP indicate loci sharing a > MP indicate groupings of loci sharing a
(The word groupings will help clarify that you mean a set, instead of just a number of other primers in a multiplex.)

Experimental design

This was research was well thought out and executed. The paper clearly addresses an important gap in knowledge, while also furthering potential resources for studying the species in the future.
The only piece of analyses I would have like to see if the split between the two populations is maintained using different grouping of the loci analyzed, to see if the structure results are qualitatively changed. I feel they should start with their complete data set and remove one locus, then rerun structure. Start with the complete data set again, then remove a different locus and rerun structure. If they repeated this process with all loci, they could strengthen their argument by demonstrating that the null alleles are not likely being overshadowed by a strongly geographically differentiated locus. They could further strengthen their argument by running structure with only the loci that had null allele levels above those traditionally accepted. Would that grouping still provide the correct structure qualitatively?
I do not think these additional steps need to have their own figures or tables, unless something qualitatively unique was found. I mainly want their analyses to show that they took those steps and give an overview of what they found.

Validity of the findings

More could be done in the introduction to describe why the population genetics of this species should be analyzed. However, I feel they demonstrated the importance of the species, posed a novel question, reported interesting results, obtained through a well thought out experiment. I greatly appreciated that their speculation was clearly identified and used as a suggestion for potential future studies.

Additional comments

I enjoyed reading your research, especially because I am also finding null alleles in my mussel species. I know my comments are numerous, but I think they will strengthen the paper. Good luck and season’s best.

·

Basic reporting

This report describes a microsatellite survey in the wedge clam Donax, and notes the common occurrence of heterozygote deficiencies relative to Hardy-Weinberg expectations. This phenomenon has been observed in a variety of studies of bivalves, in genetic markers ranging from allozymes to SNPs, and has prompted much discussion over the decades. In the case of DNA markers, as the authors observe, a favored explanation is one of PCR priming site polymorphism, which results in amplification failure. However, other explanations are possible but not mentioned, such as segmental aneuploidy, where one chromosome has a deletion containing the primer binding site. In a study of allelic BAC sequences in the Pacific oyster, for example, 42 of 101 microsatellite loci were present in the hemizygous state, owing to indels of various sizes (Oyster Genome Consortium, unpubl. data). In any case, the phenomenon appears to be common, regardless of mechanism. This paper is concerned with the possibility that high levels of null alleles can lead to a biased view of population structure.
I wasn’t sure what the authors meant by “only 16 loci produced codominant fragments”, since without inheritance data, one can only assume codominance from fragment frequencies. Maybe this could be explained further.
The hypothesis that allele size variation may contribute to allele dropout is interesting, and testable: longer alleles should occur as apparent homozygotes more often than shorter alleles. Whether the difference of a few base pairs can result in allele dropout during PCR is a good question, and may have been addressed in the microsatellite literature.
The authors suggest that NGS provides a new way of generating large numbers of microsatellite markers, but at the same time this study and others point to the technical difficulties involved in getting reliable data from this class of markers. The production of SNP markers by NGS is an attractive alternative.
At the end of the Discussion section, the authors refer to null alleles arising from mismatches in the 3’ end of priming sequences, but the data provided do not give any information on this question. Other potential causes of amplification failure exist, as noted above.
Although high frequencies of null alleles were noted, it’s not clear that they render the loci “completely useless”. The paper by Chapuis & Estoup (2007) noted the bias in Fst estimation, but provided correction methods. I think the case may have been overstated here.
I don’t follow the logic of the final argument (that the large heterozygote deficits found cannot be due solely to null alleles, since global population structure was detected). It may be the case that the data yielded a biased estimate of Fst, but the pattern of geographic differentiation does not shed much light on the source of heterozygote deficits.

Experimental design

Good effort to probe the possible artifacts in microsatellite development and scoring.

Validity of the findings

As discussed above, some of the conclusions seem unjustified by the data.

Additional comments

It would be good to revisit the interpretation of your findings, and dig a little deeper into the possible causes of nonamplifying microsatellite alleles. Also, recognize that a population survey, without corresponding genomic data, has limited power of inference.

---

## Round 0.2 · accepted · Accept

I am sorry for the delay, but the referees have not responded to my request to look at your revised manuscript, and so I am responding on my own. I am satisfied with your revisions and responses to the referees, and so I see no reason to delay your manuscript any further. I am happy to move it forward to production.